# Time to End-of-Life of Patients Starting Specialised Palliative Care in Denmark: A Descriptive Register-Based Cohort Study

**DOI:** 10.3390/ijerph192013017

**Published:** 2022-10-11

**Authors:** Camilla Jøhnk, Helene Holm Laigaard, Andreas Kristian Pedersen, Eithne Hayes Bauer, Frans Brandt, Georg Bollig, Donna Lykke Wolff

**Affiliations:** 1Department of Internal Medicine, Hospital Sønderjylland, University Hospital of Southern Denmark, Sydvang 1, 6400 Sønderborg, Denmark; 2Department of Regional Health Research, University of Southern Denmark, J. B. Winsløws Vej 19, 5000 Odense, Denmark; 3Department of Clinical Research, Hospital Sønderjylland, University Hospital of Southern Denmark, Kresten Philipsens Vej 15, 6200 Aabenraa, Denmark; 4Internal Medicine Research Unit, Hospital Sønderjylland, University Hospital of Southern Denmark, Kresten Philipsens Vej 15, 6200 Aabenraa, Denmark; 5Department of Anesthesiology, Intensive Care, Palliative Medicine and Pain Therapy, HELIOS Klinikum, 24837 Schleswig, Germany

**Keywords:** Denmark, epidemiology, palliative care, palliative medicine, duration of treatment, end-of-life

## Abstract

Increasing numbers of patients are being referred to specialised palliative care (SPC) which, in order to be beneficial, is recommended to last more than three months. This cohort study aimed to describe time to end-of-life after initiating SPC treatment and to explore potential regional variations. We used national register data from all Danish hospital SPC teams. We included patients who started SPC treatment from 2015–2018 to explore if time to end-of-life was longer than three months. Descriptive statistics were used to summarise the data and a generalised linear model was used to assess variations among the five Danish regions. A total of 27,724 patients were included, of whom 36.7% (95% CI 36.2–37.1%) had over three months to end-of-life. In the Capital Region of Denmark, 40.1% (95% CI 39.0–41.3%) had over three months to end-of-life versus 32.5% (95% CI 30.9–34.0%) in North Denmark Region. We conclude that most patients live for a shorter period of time than the recommended three months after initiating SPC treatment. This is neither optimal for patient care, nor the healthcare system. A geographical variation between regions was shown indicating different practices, patient groups or resources. These results warrant further investigation to promote optimal SPC treatment.

## 1. Introduction

Palliative care aims to improve the quality of life of patients with life-threatening illnesses, including both cancer and non-cancer diagnoses [1]. The delivery of palliative care varies from country to country in relation to roles, responsibility and implementation [2]. In Denmark, palliative care is categorised as either non-specialised palliative care which most often occurs in a primary care setting, or specialised palliative care (SPC), which is treatment at a hospice, a specialised hospital unit, or from a specialised palliative care team that provide SPC in patients’ homes [3]. Only complex and severe symptoms, which cannot be resolved through non-specialised palliative care, will need a referral to SPC, and limited capacity necessitates prioritisation. The most complicated patients are prioritised and take up most resources [4].

An increasing number of patients require palliative care due to the increased prevalence of chronic life-limiting illnesses and advances in the treatment of cancer [5,6,7,8]. This poses organisational challenges in healthcare sectors with limited resources.

Previous studies and healthcare policies suggest that for palliative care to be beneficial to patients it should last for at least three to four months [9,10,11,12]. Palliative care for at least three months before death is associated with improvement in certain symptoms, improved end-of-life care, fewer hospitalisations, fewer emergency room visits, and fewer hospital deaths [9,10,13]. However, discouraging results from a recent systematic review and meta-analysis including 169 studies found that the median survival time from referral to SPC until death varied from 19 to 66 days between studies [11]. 

Research regarding early palliative care has led to a new guideline in 2016 from the American Society of Oncology recommending that patients with advanced cancer should receive palliative care within eight weeks of diagnosis [6]. Despite knowledge on the recommended length of palliative care and the benefits of adopting early palliative care, a gap might exist between recommendations and current practice, which is of concern and scientific interest. 

One factor suspected of influencing access to SPC, is patients’ place of residence. Studies have found geographical variation in the access to, and duration of, palliative care within countries [14,15,16,17]. In Denmark, a geographical variation was seen in the proportion of patients receiving SPC before death [18]. On a regional level, the variation of the proportion of patients with cancer receiving SPC before dying or becoming too ill to benefit from SPC was 77% to 88% between regions in 2016 [18], while in 2020 these variations were 79% to 86% [19]. A German study in 2015 (*n* = 7144) reported substantial geographical variations between states where the median duration varied from 18 to 32 days [20]. 

To our knowledge, no study has investigated geographical variations in Denmark of time to end-of-life after patients initiated SPC treatment. 

The aim of this study was to describe the duration of time to end-of-life of patients initiating SPC treatment between 2015 and 2018 in Denmark and to explore possible geographical variations.

## 2. Materials and Methods

### 2.1. Study Population

This nationwide register-based study using historic data included adults (aged 18 years or older) who started treatment in an SPC team in Denmark between 1 January 2015, and 31 December 2018 (Figure 1). We included patients from all SPC teams in Denmark in the following settings: inpatient treatment at hospitals, outpatient treatment at hospitals, and SPC treatment in the home. Patients in hospices were not included in the study in order to make the results comparable with other countries where hospice care is not always provided by SPC teams. Patients who were referred but did not start palliative care treatment (*n* = 49) were excluded. Patients with more than one palliative care treatment record (*n* = 886) were included with their first entry. 

### 2.2. Study Design and Setting

The Danish National Health Service provides access to tax-financed healthcare for all Danish residents. Therefore, hospital care, including SPC, does not incur any financial cost for Danish patients. Danish hospitals report information on inpatient and outpatient contacts electronically to the National Patient Register (NPR), including SPC contact [21]. 

The geographical investigation was based on 25 SPC teams and their respective palliative care units located in the five Danish regions (Capital Region of Denmark, Region Zealand, Region of Southern Denmark, Central Denmark Region and North Denmark Region). All SPC teams included patients from 2015 to 2018 except for two units. One unit only included patients in 2017 and 2018. Another SPC unit only included patients in 2015. However, both units included few patients contributing with less than 5% of patients in the associated region. 

### 2.3. Data Sources

The Danish Palliative Database (DPD) was used to extract the study population. In Denmark, all hospital patient contacts are recorded in NPR [22]. All patients registered with a contact to an SPC team in the NPR are added to the DPD when the team confirms the contact [23,24]. In addition, data about age, sex, and diagnoses are transferred to DPD from the NPR. As the hospitals depend on delivering reports to the NPR for reimbursement purposes, the registry holds highly valid longitudinal data [21]. Date of death is collected by the DPD from the Danish Civil Registration Database [23,24]. 

### 2.4. Variables

Time to end-of-life was defined as the number of days between the date of the first SPC contact and the date of death. We followed all patients until death or for a maximum of one year (365 days). For patients who died on the same date as their first contact date, the time to end-of-life was defined as one day. For descriptive purposes, we grouped time to end-of-life into three categories: short (<3 months) corresponding to a lifespan shorter than the recommended treatment period; medium (3–12 months) corresponding to the recommended treatment period, and long (1+ year) corresponding to the recommended treatment period but also so long that it might pose a challenge for the healthcare system. We also presented time to end-of-life as a median with an interquartile range so that it was possible to compare our results with other studies that have used this measurement. Furthermore, we described the duration of time from referral to an SPC team to the first contact (0–10 days or 11+ days). We used information about the patients’ age (median and three age groups; 18–30 years, 31–64 years, and 65 years or older), sex, whether or not patients had a cancer diagnosis, region of residence, and the year SPC treatment was initiated (2015, 2016, 2017, 2018). 

### 2.5. Statistical Analysis

We used descriptive statistics to summarise the data and to assess the reproducibility of our sample. We reported categorical outcomes as numbers and percentages, and non-categorical outcomes as median and interquartile range (IQR). Demographics and duration from referral to start of the SPC contact were presented as totals and stratified according to the year when treatment was initiated. Furthermore, time to end-of-life in days was presented as median with IQR. Additionally, time to end-of-life was calculated annually for each region to describe potential geographical variation.

To analyse our primary outcome of time to end-of-life, we used survival analysis. The Cox proportional hazard model has been shown to have difficulties in relation to causal inference [25]. Therefore, we used Kaplan Meier curves and pseudo-observation defined as the probability of living at three months. In order to draw inference, generalised linear model with bootstrapped confidence intervals (CIs) was fitted where the link function was set to the natural logarithm [26]. From the model, the marginal percentages and CIs for each region and year were estimated together with the overall percentage and CI across regions and years. As the estimation methods for the CI and p-value were either estimated by way of bootstrap or the delta method, model control in relation to the normality assumption was not required.

For sensitivity purposes, we also investigated whether there were regional and yearly variations again by way of the generalised linear model, and the E-value was calculated to assess how sensitive these results were in relation to unmeasured confounding. 

Analysis was conducted using Stata version 17 (StataCorp, College Station, TX, USA).

## 3. Results

### 3.1. Study Population

A total of 27,724 patients fulfilled the inclusion criteria (Figure 1). Of the included patients, 51.5% were male, the median age was 71 years (IQR 62–78 years), and the majority (94.5%) were referred with a cancer diagnosis. Demographic data are shown in Table 1. Figure 2 shows time to end-of-life according to the five Danish regions in 2015–2018. 

### 3.2. Time to End-of-Life

Figure 3 shows time to end-of-life for patients initiating SPC treatment. The proportion of patients who lived longer than three months to end-of-life was 36.7% (95% CI 36.2–37.1%) (Table 2). There was not a significant statistical variation between the four years of the study period. The proportion of patients with a long (1+ year) time to end-of-life was 11.6%.

### 3.3. Geographical Variation

The proportion of patients still alive three months after initiating SPC-treatment varied from 32.5% (95% CI 30.9–34.0%) in North Denmark Region to 40.1% (95% CI 39.0–41.3%) in the Capital Region of Denmark (*p* < 0.05) (Table 2). The e-values for the effect estimates for Region Zealand and Region of Southern Denmark show moderate robustness against unmeasured confounding factors, whilst the sensitivity lies higher for the estimates concerning Central Denmark Region and North Denmark Region, supporting considerable robustness of the association for these regions. A broader variation between time to end-of-life was found between the individual SPC teams (data not shown). 

## 4. Discussion

This national study showed that only around 37% of patients who started SPC treatment had a time to end-of-life of more than three months. The proportion of patients with more than three months in time to end-of-life varied between regions in Denmark from 33–40%. While most patients died within the first three months of initiating SPC treatment, a total of 12% of patients were still alive after the end of follow-up (1+ year). 

### 4.1. Prior Studies

#### 4.1.1. Time to End-of-Life

Despite the importance of commencing palliative care at least three months prior to death [9,10,11,12], only few studies have been conducted in this field. A Danish study from 2017 investigated the median time from referral to SPC until death using the same database as we did, but observed patients referred from 2010 to 2012 [23]. They found a median time from referral to death of 29 days [23]. Our study population was comparable according to median age and distribution of sex. However, they included hospice patients and excluded non-cancer patients, which might explain why their median is considerably lower than the 55 days found in our study [23]. Furthermore, their definition of duration is time from referral to death which prolongs the duration compared to our definition, which is time from first contact to death. A systematic review and meta-analysis from 2020 interpreting data from 23 countries, found a great variation in median survival time from SPC referral to death—from 19 days in a Chinese study to 66 days in an Egyptian study. The review included palliative patients from different settings e.g., SPC teams, community/home, and hospice [11]. In a German study, researchers found that 78% of the population died within the first two months of SPC treatment [20]. In comparison, we found that 61.8% (2018) of the population died within three months after initiating SPC. 

When investigating trends in time to end-of-life among patients in SPC treatment, we found that the literature was sparse and yielded only one study on the topic [27]. In the aforementioned study, the authors observed a lengthening of the time between referral and until death, and an increase in the proportion of patients with an early referral to SPC (more than three months before death) when comparing patients referred for SPC in 2017–2019 to patients referred for SPC in 2007 [27]. Our findings suggested a potential tendency towards an increasing number of patients receiving longer treatment. However, a longer study period is required to confirm this finding. To our knowledge, no other studies examined time trends in the duration of SPC treatment. 

#### 4.1.2. Geographical Variation

Our study demonstrated a noteworthy geographical variation in the time to end-of-life for patients initiating SPC treatment. Geographical variations were also found in other studies [15,16,17]. A German study found variation where the median duration varied from 18 (IQR 7–54) to 32 days (IQR 11–69) in different parts of the country [20]. A Canadian study from 2012 investigating end-of-life care for patients with colorectal cancer found that indicators of poor-quality end-of-life care were mostly influenced by geography. Living in rural areas and longer distance to the nearest palliative care unit were associated with reduced chances of being registered with a palliative care program [14].

### 4.2. Strengths and Limitations

Our study has several strengths. Firstly, the study was based on nationwide register data, ensuring a large SPC population (*n* = 27,724). Secondly, it has been mandatory since 2010 for SPC teams to ensure that information on all referred patients is registered in the DPD, leading to very high completeness of the database [18,19]. This minimises the risk of selection bias when selecting SPC patients. Thirdly, we analysed data over four years, making it possible to follow the development of SPC in Denmark. 

However, the study also has limitations. Firstly, analysing data throughout an even longer study period would have strengthened the results in time-trend. Secondly, we used the first day of contact to measure the time to end-of-life. Some patients were referred several days before this date and, therefore, the referral date could have been chosen as the start date. Finally, a limitation to this study may be the connection between time to end-of-life and duration of SPC treatment. Patients can conclude SPC treatment alive which was not taken into account in the analysis of data, as this information was only registered in 2015 and 2016. This could create uncertainty, especially regarding patients with a long (1+ year) time to end-of-life, as these patients are alive but may be concluded from treatment. This uncertainty may also apply for the median; however, we do not have data to support this.

We believe that the results from this study could be generalised to other countries with similar healthcare systems. This study needs to be reproduced by studies in other countries in the future to support the findings due to the paucity of the literature on the topic. 

### 4.3. Clinical Implications

Studies suggest that the optimal length of SPC treatment should be at least three to four months to maximise potential benefits for the patients [9,10,11,12]. In this study, we found that in all four years studied, only one in three patients who initiated SPC treatment lived over three months from the start of SPC treatment. This indicates late referral to SPC and underlines that, contrary to recommendations [6], early palliative care in many cases is not initiated. Barriers for this may include patient-related barriers, barriers related to the referring physician, and/or SPC team resources. A recent systematic review found that the most common patient-related barriers included stereotyping of palliative care, denial of the terminal nature of the disease, and logistical barriers (e.g., transportation problems, waiting time, and lack of palliative care providers in the community) [28]. Barriers relating to the referring physicians may include a lack of awareness of the possibilities for referral to SPC teams or that clinicians generally overestimate their patients’ survival time. In 2016, a systematic review showed that in 13 out of 18 studies clinicians overestimated patients’ survival time [29]. Similarly, in a more recent study, clinicians were accurate for slightly more than four of ten patients only [30]. Prognostication in advanced cancer is complicated and should, therefore, be based on standardised tools and standardised prognostic terminology to enhance communication about expected treatment outcomes and bodily changes with patients and caregivers [31]. Lastly, barriers related to SPC teams include capacity limitations, which can lead to a long time from referral to treatment initiation. This study showed that 66.8% of patients started within 10 days of referral. Visitation to SPC includes the team’s interpretation of the patient’s need for SPC, which means that patients with the most severe diseases and the most complex and heaviest symptom burdens are prioritised [4]. Patients with a short survival time probably have the most complex needs, which might increase the percentage of patients with a short time to end-of-life. The short time to end-of-life demonstrated in this study coupled with the above-mentioned barriers indicates a need for greater focus on timing the initiation of SPC treatment and developing initiatives to reduce barriers to initiation of SPC treatment.

While a very short time to end-of-life can have a negative impact on patient treatment, a long time to end-of-life is a challenge for SPC organisation due to limited capacity. This study showed that one in ten patients who initiated SPC treatment had a long-term (1+ year) time to end-of-life with a slight increase over the four years studied. Resources used for long-term patients may pose a challenge in accommodating newly referred patients by the SPC team within a timeframe that would be beneficial to patients. One explanation for long-term treatment could be that improved symptom relief leads to prolongation of life. In a randomised controlled trial from 2015, Bakitas et al. found a 15% improvement in one-year survival in patients with advanced cancer receiving early palliative care concomitantly with active oncological treatment compared to patients randomised to three months delayed palliative care [32].

As mentioned in the study limitations, we do not know whether the patients with 1+ year from SPC initiation to end-of-life are, in fact, in active palliative treatment or have concluded palliative treatment. Future studies should aim to investigate whether particular patient groups require long-term SPC care and how new organisational structures may manage these patients.

We found a noteworthy geographical variation in time to end-of-life between SPC teams. Among patients with more than three months to end-of-life when initiating SPC, there was a difference of seven percentage points between the Capital Region of Denmark (40%) compared to patients in North Denmark Region (33%), and five percentage points between Region Zealand (38%) and North Denmark Region.

Capital Region encompasses one of the most specialised hospitals in Denmark, which could explain some of the variation, as the percentage of patients in long-term treatment from this hospital’s SPC team is notably different from the other teams. However, we do not have specific explanations for other geographical variations between regions. One explanation could be the difference between living in a rural and an urban area of the country. Studies show that patients living in rural areas and patients living in lower-income neighbourhoods are less likely to be referred to palliative care in comparison with patients living in urban areas [33,34]. Furthermore, Adsersen et al. suggest that part of the explanation for the Danish regional variation could be related to inequity in capacity [23].

Geographical, organisational, and socioeconomic factors may have an influence on treatment duration. However, reasons for this remain somewhat unclear and should be addressed in future studies. Further research should also focus on the reasons for long-time SPC and whether there is a connection to specific patient groups and diagnoses or not.

## 5. Conclusions

In this nationwide study, we described the time to end-of-life among patients initiating specialised palliative treatment in Danish SPC teams from 2015 to 2018. In this period, most patients had a short (<3 months) time to end-of-life. Between 2015 and 2018, only one in three patients initiating SPC treatment had a time to end-of-life over three months. According to most literature, SPC treatment should last more than three months to be beneficial to patients. Variations were seen among the five Danish regions indicating different regional practices, resources, or patient groups.

SPC is a field where resources are limited and the demand for palliative care is growing. The findings of this study call for an awareness of the gap between time to end-of-life and optimal SPC treatment duration which warrants further investigation to promote optimal patient treatment and hospital resource planning.

## Figures and Tables

**Figure 1 ijerph-19-13017-f001:**
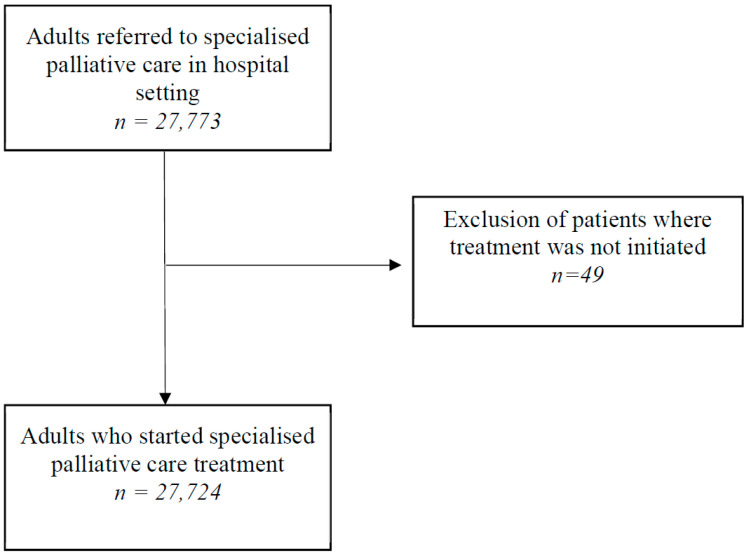
Diagram of patient selection.

**Figure 2 ijerph-19-13017-f002:**
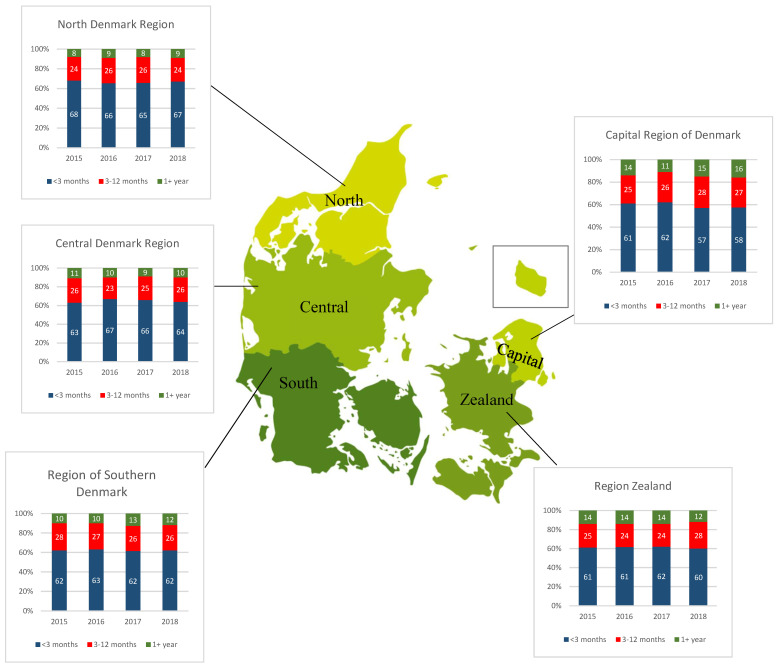
Percentages of short, medium, or long time to end-of-life by region in Denmark. Note: Time to end-of life: Short (<3 months), Medium (3–12 months), Long (1+ year).

**Figure 3 ijerph-19-13017-f003:**
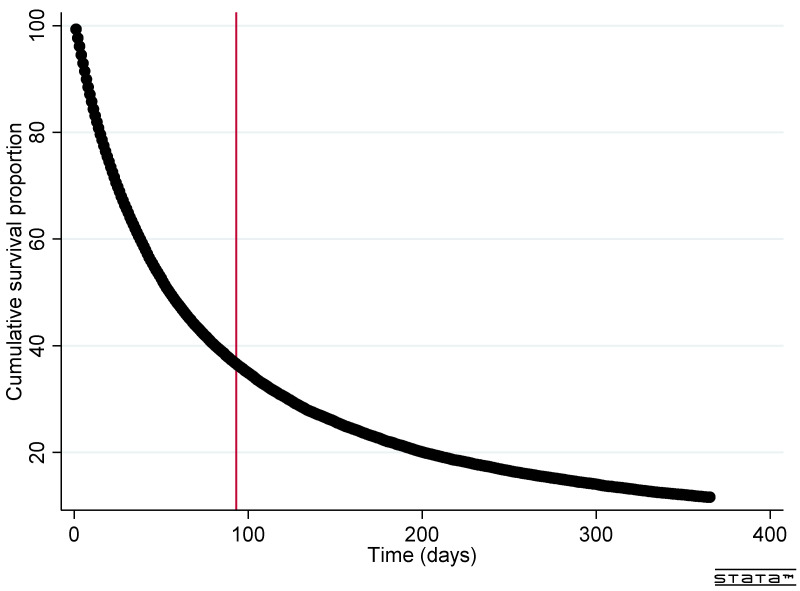
Survival curve for patients initiating SPC treatment.

**Table 1 ijerph-19-13017-t001:** Demographics of patients in specialised palliative care treatment.

	Total	2015	2016	2017	2018
	*n* (%)	*n* (%)	*n* (%)	*n* (%)	*n* (%)
	27,724 (100)	6676 (100)	7051 (100)	6944 (100)	7053 (100)
Sex					
Female	13,443 (48.5)	3240 (48.5)	3427 (48.6)	3395 (48.9)	3381 (47.9)
Male	14,281 (51.5)	3436 (51.5)	3624 (51.4)	3549 (51.1)	3672 (52.1)
Age group					
18–30	159 (0.6)	36 (0.5)	41 (0.6)	35 (0.5)	47 (0.7)
31–65	9084 (32.8)	2307 (34.6)	2317 (32.9)	2254 (32.5)	2206 (31.3)
65+	18,481 (66.7)	4333 (64.9)	4693 (66.6)	4655 (67.0)	4800 (68.1)
Referred with cancer diagnosis					
No	1532 (5.5)	218 (3.3)	291 (4.1)	411 (5.9)	612 (8.7)
Yes	26,192 (94.5)	6458 (96.7)	6760 (95.9)	6533 (94.1)	6441 (91.3)
Duration from referral to start					
0–10 days	18,507 (66.8)	4649 (69.6)	4862 (69.0)	4573 (65.9)	4423 (62.7)
≥11 days	9217 (33.2)	2027 (30.4)	2189 (31.0)	2371 (34.1)	2630 (37.3)
Time to end-of-life					
<3 months	17,355 (62.6%)	4199 (62.9%)	4489 (63.7%)	4317 (62.2%)	4350 (61.7%)
3–12 months	7149 (25.8%)	1720 (25.8%)	1784 (25.3%)	1799 (25.9%)	1846 (26.2%)
1+ year	3220 (11.6%)	757 (11.3%)	778 (11.0%)	828 (11.9%)	857 (12.2%)
Median duration (IQR)	55 (20–156)	55 (20–155)	54 (20–152)	55 (20–158)	57 (21–158)

**Table 2 ijerph-19-13017-t002:** Proportion of patients with three months to end-of-life, in total and by region and year.

	% (95% CI)	RR (95% CI)	E-Value
Total	36.7 (36.2–37.1)	n.a	n.a
Region			
Capital Region of Denmark	40.1 (39.0–41.3)	1	1
Region Zealand	38.0 (36.8–39.2)	0.95 (0.91–0.99)	1.30
Region of Southern Denmark	37.2 (36.0–38.5)	0.93 (0.89–0.97)	1.37
Central Denmark Region	34.5 (33.3–35.7)	0.86 (0.82–0.90)	1.60
North Denmark Region	32.5 (30.9–34.0)	0.81 (0.77–0.85)	1.78
Year			
2015	36.3 (35.0–37.6)	1	1
2016	35.7 (34.5–37.0)	0.98 (0.94–1.03)	1.15
2017	37.0 (36.0–38.1)	1.02 (0.97–1.07)	1.16
2018	37.5 (36.3–38.6)	1.03 (0.98–1.08)	1.21

RR: Rate Ratio. n.a: Not applicable.

## Data Availability

The data that support the findings of this study are available from Danish Palliative Database but restrictions apply to the availability of these data, which were used under license for the current study, and, therefore, are not publicly available. Data are, however, available from the authors upon reasonable request and with permission of Danish Palliative Database.

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
