# Peer review of "Time to End-of-Life of Patients Starting Specialised Palliative Care in Denmark: A Descriptive Register-Based Cohort Study"

_ijerph, 2022, doi:10.3390/ijerph192013017_

Round 1
Reviewer 1 Report
Minor comment
1. the delivery system of specialized palliative care, what kind of palliative care is provided to what kind of patients, and the sharing of roles with non-specialized palliative care varies from country to country. As such, the information should be added at the beginning of the "Introduction" section.
The content of "4.1.1 Time to end of life" is unclear in its rationale. It would be desirable to describe it based on the perspective of how it is for the Danish citizens, rather than merely comparing it with other countries.
As the purpose of the study is to clarify regional variations, it would be better to put 4.1.2. first in the order of description.
Reviewer 2 Report
Thanks for the opportunity reviewing this paper. Appropriate specialised palliative care (SPC) is important for the patients and their loved ones, so the service quality matters which includes but not limited to the access criteria, timing, duration (time to end-of-life since the first SPC contact or referral) and what proper service was provided to the patients. This papers looks at one component of the service quality, which is the duration from the first SPC contact to end-of-life. Closely related to the duration is who (in terms of the stage/seriousness/severity/complications of the diseases, and patient characteristics) have used the SPC service, together with the factors related to the service providers (resources, capacity and capability). It's good the authors have had a good discussion about the latter.
There are a few queries for the authors to clarify, as follows:
1 There were 2 teams with incomplete data (Line 90). I wonder why it happened and how likely if it might influence the findings.
2 The findings were shown as either by year (all regions combined) or by region (all years combined for a particular region). I wonder if there was evidence that there was interaction between region and year, e.g. year modified the effect of region.
3 The first SPC contact was used as the start of the duration to end-of-life. In this study, how many patients started treatment soon after the referral and before the first SPC contact? A proportion would be helpful so that readers can appreciate the degree of the underestimate of the duration in days (and the proportion of less than 3 months).
4 Why hospice patients were not included in this study?
5 The resolution of Figure 2 needs to be improved.
6 The statistical methods need to be further explained and referred: a reference for generalised linear model with bootstrapped confidence intervals (CIs) would be helpful. In addition, interpretation of the E-value will benefit the readers.
Spellings and other minor things as follows:
Line 63: the variation (of the proportions) of the patients: please include the words in brackets.
Line 89: please add the names of the five Danish regions in body text.
Line 120: ... start (of the SPC contact)...: please include the words in brackets.
Line 121: (when) treatment was initiated: please include the word in brackets.
Table 1: ">= 11 days" rather than "11 <=days".
Table 1: "<3 months" rather than "0-3 months" (as it's unclear if "3" was included. Similarly, please use "3- months" rather than "3-12 months".
Table 2: please give the meaning of 'RR' in table footnote (rate ratio).
Figure 3: please re-label the two axes. The horizontal axis should be labelled with "Time (days)". I think the vertical one should be cumulative survival probability.
Lines 174-179: another difference between yours and the 2017 Danish study was the methods used for the calculation for the duration: one from the referral to death and the other from the first SPC contact to end-of-life.
Line 153 and Lines 192/193: can you clarify which is true or if you can better re-word these lines?
Line 200: The British study was based on patients in hospices - I'm convinced it's a good comparison for reference.
Line 289: "Both in 2015 and 2018" or "Between 2015 and 2018"?
Reviewer 3 Report
This is an interesting study on variation between regions in Denmark on when patients are referred to specialized palliative care in relation to remaining survival time. Although the result might primarly be of importance for Danish palliative care the results might also be of interest for a wider readership. The study discuss the differances in referral of patients to palliative care which highlights the importance of identifiaction of palliative care patiente in time and about prognostication. I think the study is well-written with an impressive large cohort and solid methods.
My only concern is the lack of information about the socioeconomic status in the different geographical regions. Might this explain/contribute to the differences reported? Could the authors add some information about e.g. the average income in the different regions.
I also think that it would be of interest to add some more discussion about prognosticiation in palliative care.
Round 2
Reviewer 3 Report
All my queries have been adressed properly.